# Evaluation of MicroRNAs as Non-Invasive Diagnostic Markers in Urinary Cells from Patients with Suspected Prostate Cancer

**DOI:** 10.3390/diagnostics10080578

**Published:** 2020-08-09

**Authors:** Angelika Borkowetz, Andrea Lohse-Fischer, Jana Scholze, Ulrike Lotzkat, Christian Thomas, Manfred P. Wirth, Susanne Fuessel, Kati Erdmann

**Affiliations:** 1Department of Urology, Technische Universität Dresden, 01307 Dresden, Germany; angelika.borkowetz@uniklinikum-dresden.de (A.B.); andrea.lohse-fischer@uniklinikum-dresden.de (A.L.-F.); jana.scholze@uniklinikum-dresden.de (J.S.); ulrike.lotzkat@uniklinikum-dresden.de (U.L.); christian.thomas@uniklinikum-dresden.de (C.T.); manfred.wirth@uniklinikum-dresden.de (M.P.W.); susanne.fuessel@uniklinikum-dresden.de (S.F.); 2German Cancer Consortium (DKTK), Partner Site Dresden, 01307 Dresden and German Cancer Research Center (DKFZ), 69120 Heidelberg, Germany; 3National Center for Tumor Diseases (NCT), 01307 Dresden, Germany: German Cancer Research Center (DKFZ), Heidelberg, Germany; Faculty of Medicine and University Hospital Carl Gustav Carus, Technische Universität Dresden, Dresden, Germany; Helmholtz-Zentrum Dresden-Rossendorf (HZDR), Dresden, Germany

**Keywords:** biomarker, biopsy, diagnosis, microRNA, non-invasive, prostate cancer, PSA, PSA density, urinary cells, urine

## Abstract

Currently used tumor markers for early diagnosis of prostate cancer (PCa) are often lacking sufficient specificity and sensitivity. Therefore, the diagnostic potential of selected microRNAs in comparison to serum PSA levels and PSA density (PSAD) was explored. A panel of 12 PCa-associated microRNAs was quantified by qPCR in urinary sediments from 50 patients with suspected PCa undergoing prostate biopsy, whereupon PCa was detected in 26 patients. Receiver operating characteristic (ROC) curve analyses revealed a potential for non-invasive urine-based PCa detection for miR-16 (AUC = 0.744, *p* = 0.012; accuracy = 76%) and miR-195 (AUC = 0.729, *p* = 0.017; accuracy = 70%). While serum PSA showed an insufficient diagnostic value (AUC = 0.564, *p* = 0.656; accuracy = 50%) in the present cohort, PSAD displayed an adequate diagnostic performance (AUC = 0.708, *p* = 0.031; accuracy = 70%). Noteworthy, the combination of PSAD with the best candidates miR-16 and miR-195 either individually or simultaneously improved the diagnostic power (AUC = 0.801–0.849, *p* < 0.05; accuracy = 76–90%). In the sub-group of patients with PSA ≤ 10 ng/mL (*n* = 34), an inadequate diagnostic power of PSAD alone (AUC = 0.595, *p* = 0.524; accuracy = 68%) was markedly surpassed by miR-16 and miR-195 individually as well as by their combination with PSAD (AUC = 0.772–0.882, *p* < 0.05; accuracy = 74–85%). These findings further highlight the potential of urinary microRNAs as molecular markers with high clinical performance. Overall, these results need to be validated in a larger patient cohort.

## 1. Introduction

In 2018, prostate cancer (PCa) accounted for about 1.3 million newly diagnosed cases and 360,000 deaths and thus, was the second most frequent tumor and the fifth-leading cause of cancer-related deaths among men worldwide [1]. Consequently, the management of PCa contributes significantly to overall health care costs [2]. PCa is usually suspected on the basis of an abnormal digital-rectal examination (DRE) and/or an elevated serum level of the prostate-specific antigen (PSA) [3]. Subsequently, a prostate biopsy is performed to obtain tissue samples for the histopathological verification. While PSA testing is associated with good sensitivity, it displays a low specificity due to the fact that PSA levels can also be elevated in men with benign prostatic hyperplasia (BPH) or prostatitis [4]. PSA testing also results in a higher incidence of low-risk PCa, which do not require treatment, and thus, leads to an increased number of unnecessary prostate biopsies and subsequent overtreatment [3,4]. Depending on the biopsy technique, side effects can include infectious complications to the point of life-threatening sepsis, rectal pain, rectal or perineal bleeding, hematuria, hematospermia, transient erectile dysfunction, or urine retention [3,5].

In order to increase the specificity of PSA testing, various PSA derivatives, such as free PSA, proPSA, PSA density (PSAD), and PSA velocity, have been implemented [4]. PSAD is calculated as the quotient of the serum PSA level and the prostate volume and shows an improvement of diagnostic accuracy over PSA, but its use for early PCa diagnosis is to date not commonly recommended in guidelines [6]. Established PSAD cutoffs are 0.1 or 0.15 ng/mL^2^, which, however, may lead to underdetection of significant PCa [6]. In addition, PSAD depends on the technique used for prostate size measurement and shows high interexaminer variability [4]. Therefore, additional biomarkers are needed to avoid or postpone unnecessary biopsies and to better identify patients with aggressive PCa.

Due to the molecular heterogeneity of PCa, the identification and clinical translation of disease- and stage-specific molecular markers is a rational approach to further advance diagnosis, which also paves the way for more personalized medicine. Promising molecular biomarkers for PCa include mRNAs, microRNAs (miRNAs), long non-coding RNAs (lncRNAs), and proteins [7,8,9]. Particularly miRNAs represent attractive biomarker candidates, because they are expressed in a tissue- and tumor-specific manner. Furthermore, miRNAs are generally stable and can be reproducibly extracted from a wide range of biologic samples, such as tissue, urine, blood, and exosomes, from various body fluids [10]. As post-transcriptional regulators of specific target mRNAs, miRNAs are involved in many physiological and pathological pathways, such as proliferation, cell cycle, apoptosis, androgen receptor signaling, stemness, etc. [11,12]. Consequently, they are implicated in cancer onset, recurrence, progression, metastasis, and therapy resistance by exerting tumor-suppressive or oncogenic effects [12].

Extensive investigations using malignant and non-malignant prostate tissue have shown that certain miRNAs are also differentially expressed in PCa [11,13,14,15,16]. Furthermore, selected miRNAs possessed diagnostic potential when analyzed in urine and/or blood samples from PCa patients and control subjects [10,11,17,18]. Particularly, the collection of urine is inexpensive, simple, and non-invasive, whereupon the number of prostatic cells and thus the amount of PCa-specific miRNAs in the urine specimens can be increased by DRE. Therefore, in the present study, the expression of 12 miRNAs commonly deregulated in PCa (miR-125b, -145, -155, -16, -195, -200c, -205, -21, -218, -26a, -375, -96) was analyzed in urinary sediments, which were obtained post-DRE and pre-biopsy from 50 patients with suspected PCa. The results revealed a potential for non-invasive urine-based PCa detection for miR-16 and miR-195 with greater accuracy than PSA and PSAD.

## 2. Materials and Methods

### 2.1. Patient Cohort

The patient cohort consisted of 50 consecutive patients with suspected PCa scheduled for prostate biopsy due to an elevated serum PSA level (≥4.0 ng/mL) and/or abnormal DRE. All patients received a transrectal ultrasound-guided systematic 12-core biopsy at the Department of Urology (Technische Universität Dresden, Germany) between June and September 2016. Forty-one patients (82%) underwent multiparametric magnetic resonance tomography (mpMRI) and in the case of tumor-suspicious lesions according to prostate imaging reporting and data system (PI-RADS ≥ 3) received additional targeted MRI/ultrasound fusion biopsy. Based on the biopsy outcome, patients were then classified into the tumor-free (Tf) or PCa group. The Gleason Scores (GSs) of the biopsy specimens were determined according to the latest ISUP grading system and then patients were classified into specific Gleason grade groups [19]. In patients with diagnosed intermediate- or high-risk PCa, the presence of lymphogenic or distant metastasis was assessed by computed tomography scan and bone scan. Post-DRE urine collection and analysis were approved by the institutional review board of the Medical Faculty at the Technische Universität Dresden (ethical approval code: EK 23022006, 28 February 2006). Written informed consent was obtained from each patient.

### 2.2. Collection and Processing of Urine Samples

After receiving a DRE and before biopsy, spontaneous urine samples (20–100 mL, 2nd or 3rd urine of the day) were collected in sterile containers from each patient. Urine samples were immediately centrifuged at 870× *g* for 5 min at 4 °C. The supernatant was discarded, and the cellular pellet was washed twice with ice-cold phosphate-buffered saline (PBS) by centrifugation at 870× *g* for 5 min at 4 °C. The remaining cellular pellet was resuspended in 700 µL of QIAzol Lysis Reagent (Qiagen, Hilden, Germany) and stored at −80 °C until further processing.

### 2.3. RNA Isolation and cDNA Synthesis

Total RNA from the lysed cellular pellets was extracted using the Direct-zol RNA MiniPrep Kit (Zymo Research, Freiburg, Germany) according to the manufacturer’s description. RNA quality and quantity were determined by means of the NanoDrop 2000c spectrophotometer (PEQLAB, Erlangen, Germany) and the Agilent RNA 6000 Pico Kit on an Agilent 2100 Bioanalyzer (Agilent Technologies, Ratingen, Germany).

Total RNA of up to 390 ng was then reverse transcribed into cDNA using the TaqMan MicroRNA Reverse Transcription Kit and Megaplex RT Primers (Human Pool A v2.1; both Thermo Fisher Scientific, Darmstadt, Germany), which allows for reverse transcription of up to 381 miRNAs and controls in a single reaction. Subsequently, an unbiased pre-amplification step using Megaplex PreAmp Primers (Human Pool A v2.1) and the TaqMan PreAmp Mastermix (both Thermo Fisher Scientific) was performed by following the manufacturer’s recommendations in order to enrich the amount of cDNA and thus to improve the quantification sensitivity described in the next chapter.

### 2.4. Quantitative Polymerase Chain Reaction

For quantification of miRNAs by quantitative polymerase chain reaction (qPCR), TaqMan microRNA Expression Assays (Thermo Fisher Scientific) were used, which contained the appropriate primers and probes: miR-125b-5p (#000449), miR-145-5p (#002278), miR-155-5p (#002623), miR-16-5p (#000391), miR-195-5p (#000494), miR-200c-3p (#002300), miR-205-5p (#000509), miR-21-5p (#000397), miR-218-5p (#000521), miR-26a-5p (#000405), miR-375-3p (#000564), miR-96-5p (#000186), RNU44 (#001094, reference), and RNU48 (#001006, reference). Each qPCR was performed in a total volume of 10 µL and contained 1 µL of pre-amplified cDNA (1:40), 0.5 µL of specific TaqMan microRNA Expression Assay, 5 µL of GoTaq Probe qPCR Master Mix (Promega, Mannheim, Germany), and 3.5 µL of nuclease-free water. Samples without cDNA templates served as negative controls. Two independent reactions were run per sample using the LightCycler 480 Real-Time PCR System (Roche Diagnostics, Mannheim, Germany) and the qPCR conditions were as follows: Initial denaturation at 95 °C for 10 min followed by 45 cycles at 95 °C for 15 s and 60 °C for 60 s. Crossing points (CPs) were determined by the automatic second derivative method and then averaged. If both CP values deviated >0.5, the measurement was repeated. Relative expression levels were then calculated by the ΔΔCP method and by normalization to the geometric CP mean of the small nucleolar RNAs RNU44 and RNU48 (geoM).

### 2.5. Statistical Analysis

Statistical analysis was carried out using GraphPad Prism Version 8.2.1 for Windows (GraphPad Software, San Diego, CA, USA). The Mann–Whitney U test was used for two group comparisons, whereas the chi-squared test was used to evaluate associations among categorized variables. Receiver operating characteristic (ROC) curve analysis and the corresponding area under the curve (AUC) values including the 95% confidence intervals (95% CIs) were used to evaluate the diagnostic potential of PSA, PSAD, and miRNAs alone as well as in various combinations. Then, the Youden index was calculated to define the optimal cutoff value for miRNAs alone or in combination and thus to classify patients into the Tf or PCa group. Based on the respective cutoff value, the sensitivity (SNS), specificity (SPC), positive predictive value (PPV), negative predictive value (NPV), positive likelihood ratio (pLR), negative likelihood ratio (nLR), and accuracy (ACC) were determined. Overall, all *p* values were corrected for multiple comparisons according to the method of Benjamini and Hochberg to control the false discovery rate [20]. An adjusted *p* value <0.05 was considered statistically significant.

## 3. Results

### 3.1. Characteristics of Patients Undergoing Prostate Biopsy

In this study, 50 patients scheduled for prostate biopsy due to suspected PCa were included, whereupon PCa was detected in 26 patients (Table 1). Nine patients (18%) received a systematic biopsy only, whereas 41 patients (82%) were subjected to additional targeted biopsy by MRI/ultrasound-fusion biopsy due to tumor-suspicious lesions in mpMRI. The median age and the median PSA level of the PCa group did not significantly differ from those of the Tf group comprised of patients with negative biopsy results. However, the median PSAD was, by trend, higher in the PCa group compared to the Tf group (*p* = 0.060). Additionally, patients with negative biopsies, per trend, less often had positive pre-biopsy DRE than patients from the PCa group (*p* = 0.093). Within the PCa group, 25 patients were diagnosed with an organ-confined PCa, whereas only one patient presented a locally advanced PCa. Furthermore, lymph node and distant metastases were only detected in one and two patients, respectively. Concerning the tumor grading, 16 patients were classified to Gleason grade groups ≤2 and 10 patients to the Gleason grade groups >3.

### 3.2. Deregulation of Urinary miRNAs in PCa

Next, 12 selected miRNAs (miR-125b, -145, -155, -16, -195, -200c, -205, -21, -218, -26a, -375, -96) were analyzed by qPCR in urinary cells, which were obtained from urine post-DRE and pre-biopsy from patients with positive and negative prostate biopsy. Of all investigated miRNAs, only miR-16 and miR-195 were significantly deregulated (*p* = 0.040; Table 2, Figure 1A,B). Both miRNAs were downregulated by about 40% in urinary sediments from patients diagnosed with PCa in comparison to the Tf group. Furthermore, miR-145 and miR-26a exhibited, per trend, a downregulation in urinary sediments from patients with positive biopsies compared to patients with negative biopsies (Table 2, Figure 1C,D). However, this difference was not significant following correction for multiple comparisons (*p* = 0.093).

Furthermore, none of the evaluated miRNAs showed a significant association with clinico-pathological parameters when patients were dichotomized according to their age (≤median vs. >median; *p* > 0.05), serum PSA level (≤10 ng/mL vs. >10 ng/mL; *p* > 0.05), and PSAD (≤0.15 ng/mL^2^ vs. >0.15 ng/mL^2^; *p* > 0.05). Within the PCa group, none of the miRNAs were associated with Gleason grade groups (≤2 vs. >3; *p* > 0.05).

In the sub-cohort of patients with serum PSA levels ≤10 ng/mL (*n* = 34), miR-16 and miR-195 were also significantly downregulated by more than 50% in the PCa group compared to the Tf group (*p* = 0.017 and 0.045; Appendix A). Furthermore, miR-145, miR-21, miR-26a, and miR-96 were also slightly but not significantly downregulated in urinary sediments from patients with positive biopsies compared to patients with negative biopsies (*p* > 0.050, Appendix A)

### 3.3. Diagnostic Potential of miRNA Expression Levels in Urinary Cells

Next, the diagnostic potential of the miRNA expression levels in urinary sediments regarding the discrimination of positive and negative prostate biopsies was investigated using the whole study cohort (*n* = 50). Furthermore, their diagnostic performance was compared to the established biomarkers PSA and PSAD. While serum PSA levels showed an insufficient diagnostic value (AUC = 0.564, 95% CI = 0.403-0.726, *p* = 0.656) in the present cohort, PSAD displayed an adequate diagnostic performance (AUC = 0.708, 95% CI = 0.562-0.853, *p* = 0.031) (Figure 2A and Appendix A). The ACC values for PSA (cutoff 4 ng/mL) and PSAD (cutoff 0.15 ng/mL^2^) were 50% and 70%, respectively.

ROC curve analyses also revealed a potential for non-invasive urine-based PCa detection for miR-16 (AUC = 0.744, 95% CI = 0.599-0.888, *p* = 0.012) and miR-195 (AUC = 0.729, 95% CI = 0.587–0.871, *p* = 0.017) (Figure 2B and Table 3). Particularly, miR-16 outranked the diagnostic performance of PSAD, exhibiting higher AUC, SPC, PPV, NPV, pLR, and ACC values as well as a lower nLR value (Table 3). Most remarkably, SPC, PPV, and ACC increased by about 17%, 13%, and 6%, respectively, compared to PSAD.

Furthermore, miR-145 and miR-26a showed a per trend diagnostic potential (each *p* = 0.070), with AUCs of 0.674 (95% CI = 0.519–0.829) and 0.678 (95% CI = 0.521–0.834), respectively (Appendix A and Appendix A). In contrast, the remaining miRNAs were without substantial diagnostic potential (Appendix A).

To test if the diagnostic performance can be further enhanced, various combinations of PSAD and expression levels of miR-16 and miR-195 were evaluated by ROC curve analyses. For this purpose, sum scores were generated according to if PSAD and miRNA expression levels predicted a positive (1) or negative (0) biopsy result. Combinations of PSAD with both miRNAs either individually or simultaneously as well as the combination of both miRNAs resulted in an elevated diagnostic power (AUC = 0.772-0.849, *p* < 0.005, ACC = 76–90%) (Figure 3 and Table 3). This was mostly accompanied by higher SNS, SPC, PPV, NPV, pLR, and ACC values as well as by lower nLR values (Table 3). The best diagnostic performance was achieved by the combination of PSAD/miR-16, with SNS, SPC, PPV, NPV, and ACC values >87% compared to PSAD alone, where these parameters exhibited values of <72%. Particularly, ACC was increased by 20% from 70% for PSAD to 90% for miR-16/PSAD. Furthermore, pLR and nLR values were highly increased and decreased, respectively.

### 3.4. Diagnostic Potential of miRNA Expression Levels in Urinary Cells in the Sub-Cohort of Patients with PSA Levels ≤ 10 ng/mL

In the next step, the capability of the investigated miRNAs to discriminate between positive and negative biopsies was evaluated in the sub-cohort of patients with PSA levels ≤10 ng/mL (*n* = 34). Here, both PSA and PSAD showed inadequate diagnostic power with AUC values of 0.621 (95% CI = 0.431–0.811, *p* = 0.379) and 0.595 (95% CI = 0.394–0.795, *p* = 0.524), respectively (Figure 4A and Appendix A). PSA (cutoff 4 ng/mL) and PSAD (cutoff 0.15 ng/mL^2^) predicted 41.2% and 67.6% of the cases correctly, respectively. Regarding the diagnostic value of miRNA expression levels in urinary sediments, miR-16 (AUC = 0.818, 95% CI = 0.659–0.976, *p* = 0.008) and miR-195 (AUC = 0.772, 95% CI = 0.614–0.930, *p* = 0.022) emerged once again as the best candidates (Figure 4B and Table 4). In detail, miR-16 exhibited the highest AUC, SPC, PPV, NPV, pLR, and ACC values as well as the lowest nLR value.

Apart from miR-16 and miR-195, miR-145 (AUC = 0.696, 95% CI = 0.514–0.879, *p* = 0.117) and miR-21 (AUC = 0.705, 95% CI = 0.522–0.888, *p* = 0.109) also showed diagnostic potential, but ROC curve analyses lost their significance after correction for multiple testing (Appendix A and Appendix A). All other miRNAs were without substantial diagnostic potential in this patient sub-cohort (Appendix A).

Furthermore, various combinations of PSAD and expression levels of miR-16 and miR-195 were evaluated by ROC curve analyses and then compared based on their diagnostic parameters. Compared to PSAD alone, all investigated combinations resulted in an elevated diagnostic power (AUC = 0.798–0.882, *p* < 0.05, ACC = 73–82%) (Figure 5 and Table 4). Particularly, the combinations miR-16/PSAD, miR-16/miR-195, and miR-16/miR-195/PSAD exhibited mostly higher SNS, SPC, PPV, NPV, pLR, and ACC values as well as lower nLR values (Table 4). Considering all diagnostic parameters, however, none of the investigated combinations could outperform miR-16 in its diagnostic performance.

## 4. Discussion

In the present study, the diagnostic potential of 12 PCa-associated miRNAs in urinary sediments was evaluated and compared to established diagnostic biomarkers PSA and PSAD with a focus on the latter. Serum PSA is the primary biomarker for early PCa detection, but its low specificity consequently results in unnecessary prostate biopsies [3,4]. In the present study cohort comprised of 26 patients with a positive biopsy and 24 patients with a negative biopsy, the pre-biopsy serum PSA level also showed an inadequate diagnostic performance (AUC = 0.564, ACC = 50%). In contrast, the PSA derivate PSAD, which is associated with a modest improvement in diagnostic accuracy [4], displayed a better diagnostic performance (AUC = 0.708, ACC = 70%) than PSA. In the sub-cohort of patients with PSA levels ≤10 ng/mL, both PSA and PSAD inadequately predicted PCa (AUC = 0.621 and 0.595, ACC = 41% and 68%).

Within the gray zone of serum PSA levels of 4–10 ng/mL, 75% of men who are subjected to ultrasound-guided systematic biopsy do not have PCa in the first biopsy [4]. During the last decade, the use of mpMRI has been introduced for primary PCa diagnosis since mpMRI of the prostate showed a higher sensitivity and specificity in the detection of clinically significant PCa [21,22]. Targeting of tumor-suspicious lesions in mpMRI by MRI/ultrasound-fusion biopsy showed a higher detection rate of significant PCa [23,24,25] and a better prediction of tumor aggressiveness [26,27]. However, mpMRI of the prostate still presents a false-negative rate of significant PCa in up to 15–20% [27,28,29]. Therefore, additional diagnostic biomarkers and imaging modalities are mandatory to ameliorate PCa diagnosis and to prevent unnecessary prostate biopsies. Promising biomarker candidates include miRNAs and lncRNAs, which can non- or minimal-invasively be determined in body fluids, such as urine or blood. The commercially available Progensa test, for instance, determines the urinary transcript level of the lncRNA prostate cancer antigen 3 (*PCA3*). Although *PCA3* might serve as a valuable biomarker for repeat prostate biopsy decision, the ideal cutoff point remains controversial [4,7,8].

Due to their tumor-specific deregulation, miRNAs also possess diagnostic potential for PCa detection [10,17,18]. In addition, miRNAs have the advantage of being generally stable in various body fluids and thus, they are deemed to be useful biomarkers [10]. In a time course experiment, Salido-Guadarrama et al. could demonstrate that the expression levels of miR-21 and miR-15a remained relatively constant in urinary cells [30]. In contrast, transcript levels of the mRNAs PSA and glyceraldehyde 3-phosphate dehydrogenase (*GAPDH*) decreased rapidly over 24 h.

Of the investigated miRNA panel, miR-16 and miR-195 were significantly downregulated by about 40% in urinary cells obtained from patients with a positive prostate biopsy compared to the Tf controls. In accordance, other studies have shown a decreased expression of both miRNAs in PCa tissues compared to non-malignant prostate tissue [14,15]. Furthermore, Salido-Guadarrama et al. demonstrated a significant downregulation of miR-195 by about 60% in urinary cells from patients with positive prostate biopsies in comparison to patients with negative prostrate biopsies [30]. In PCa, miR-16 and miR-195 have been considered as tumor-suppressive miRNAs as their overexpression resulted in decreased PCa cell growth and invasion as well as in enhanced radiosensitivity in vitro and/or in vivo [31,32,33,34,35]. In addition, high levels of miR-16 and miR-195 were significantly associated with longer biochemical recurrence-free survival of PCa patients [31,34].

Due to their differential expression, miR-16 and miR-195 alone were able to discriminate between PCa and Tf biopsies in the present cohort and showed increased diagnostic potential compared to PSAD alone. Combinations of PSAD with both miRNAs either individually or simultaneously as well as the combination of both miRNAs resulted in an even more elevated diagnostic power. The best diagnostic performance was achieved by the combination PSAD/miR-16, displaying the highest SNS, SPC, PPV, NPV, pLR, and ACC values as well as the lowest nLR value. AUC and ACC were increased from 0.708 and 70% for PSAD to 0.834 and 90% for PSAD/miR-16. Furthermore, the capability of the investigated miRNAs to discriminate between positive and negative biopsies was also evaluated in the sub-cohort of patients with PSA levels ≤10 ng/mL. Interestingly, the diagnostic power of PSAD was similar to PSA in this sub-cohort and was markedly surpassed by miR-16 and miR-195 individually as well as by their combinations. Considering all diagnostic parameters, miR-16 emerged as the best diagnostic candidate with an AUC of 0.818 and ACC of 85% compared to 0.595 and 68%, respectively, for PSAD. To the best of our knowledge, this is the first study reporting a diagnostic potential for miR-16 and miR-195 in urinary cells from patients with suspected PCa.

A systematic review by Paiva et al. including 18 primary studies suggested that miR-21, miR-141, miR-375, and miR-574 should be considered as potential urinary biomarkers for PCa diagnosis [17]. However, these 18 studies varied in methodical conditions and used different urinary fractions (whole urine, urinary cells, or exosomes) and time points of collection (post-DRE or not). Similar to our approach, only the studies by Foj et al. [36], Salido-Guadarrama et al. [30], Stephan et al. [37], and Casanova-Salas et al. [38] investigated the expression of miRNAs by qPCR in urinary sediments obtained from post-DRE and pre-biopsy urines (Table 5).

The study by Salido-Guadarrama et al. identified various deregulated miRNAs, including miR-195 and miR-200c between the PCa and Tf group, but only further evaluated the diagnostic potential of miR-100 and miR-200b [30]. A combination of miR-100 and miR-200b was proved to outperform the capability of serum PSA to predict PCa using the whole study cohort (AUC = 0.738 vs. 0.681, ACC = 75.5% vs. 67.8%) as well as the sub-group of patients in the PSA gray zone of 4–10 ng/mL (AUC = 0.827 vs. 0.590, ACC = 80.5% vs. 68.8%). A combination of the signature miR-100/miR-200b with the clinical parameters age, serum PSA, % of free PSA, and DRE further enhanced the diagnostic potential in the entire cohort (AUC = 0.876, ACC = 81.8) and in the sub-group PSA gray zone (AUC = 0.686, ACC = 84.4%). However, unlike the present work, Salido-Guadarrama et al. did not include patients with clinical insignificant PCa (GS ≤ 6) in their analysis [30].

Casanova-Salas et al. reported an independent predictive value for positive biopsy for miR-187 but not for miR-182 [38]. A prediction model including serum PSA, urinary *PCA3*, and miR-187 provided an improved diagnostic potential compared to PSA alone (AUC 0.711 vs. 0.615). Recently, Nayak et al. also investigated the expression of miR-182 and miR-187 in urinary cells derived from post-DRE and pre-biopsy urines (Table 5) [39]. However, both miRNAs were not differently expressed in the PCa and Tf group and thus, the diagnostic potential was not further evaluated.

Furthermore, Foj et al. demonstrated that upregulated miR-21, miR-141, and miR-375 levels as well as downregulated miR-214 levels in urinary cells were predictive for PCa diagnosis (AUC = 0.707–0.817) [36]. The limitation of this study was the small sample size of the Tf group, which contained urine from healthy volunteers, whereas the PCa group comprised 60 patients with biopsy-proven PCa.

A few studies also analyzed miRNA expression in urinary cells from urine samples, which were collected from patients post-biopsy (Table 5) [40,41,42]. Using post-DRE and post-biopsy urines, Bryant et al. reported a higher expression of miR-107 and miR-574-3p in urinary cells from patients with diagnosed PCa compared to Tf control subjects [40]. Both miRNAs predicted PCa more accurately than *PCA3* (AUC = 0.740 and 0.660 vs. 0.610). Notably, the sample size of the PCa and Tf group was distinctively unequal (*n* = 118 vs. 17). Within two studies, Stuopelytė et al. analyzed miRNA expression in urinary cells derived from either catheterized or voided urine from patients with diagnosed PCa and BPH before radical prostatectomy (pre-RPE) as well as voided urine from healthy men [41,42]. They demonstrated a diagnostic potential for miR-21, miR-148a, and miR-375. It should be noted that studies using post-biopsy and/or pre-RPE urine samples do not reflect the clinical situation as low-risk PCa might not be included. Additionally, an ideal biomarker should give diagnostic information before any invasive intervention and thus, should be tested in a pre-biopsy setting.

Apart from miR-16 and miR-195, miR-145, miR-26a, and miR-21 were also slightly downregulated in the PCa group and might also possess diagnostic potential when analyzing the entire cohort and/or the sub-cohort of patients with PSA levels ≤10 ng/mL. Neither miR-145 nor miR-26a have been evaluated as diagnostic markers for PCa in the aforementioned studies analyzing miRNA expression in urinary cells from post-DRE and pre-biopsy urines (Table 5). MiR-145 is frequently described to be decreased in PCa and has exhibited antiproliferative and proapoptotic functions in vitro [14,43,44]. Regarding miR-26a, we previously showed that this miRNA is significantly downregulated in PCa tissue samples and could exert tumor-suppressive functions possibly by regulating genes that are upregulated in PCa, such as alpha-methylacyl-CoA racemase (*AMACR*) and enhancer of zeste homolog 2 (*EZH2*) [13,45]. In contrast, miR-21 is considered oncogenic and known to be upregulated in PCa tissues [46,47,48]. An upregulation of miR-21 was associated with more aggressive tumor features as well as shorter biochemical recurrence-free and progression-free survival [46,47,48]. In the present study, however, miR-21 was slightly but not significantly downregulated in urinary cells from patients with detected PCa. In accordance with our study, Stuopelytė et al. also detected lower miR-21 levels in urinary cells from PCa patients, which was accompanied by a discriminatory power to separate PCa from BPH patients (AUC = 0.633) [42]. In contrast, Foj et al. found that a higher miR-21 level in urinary cells was a predictor for PCa (AUC = 0.817) [36]. However, both studies differed distinctly in their selection of study subjects: Pre-biopsy vs. pre-RPE for the PCa group and healthy men vs. men with BPH for the Tf group (Table 5) [36,42]. Another explanation for these contradictory results could be that urine is a variable composed biofluid depending on the hydration status and the condition of the urinary tract system of the subject, which in turn could influence the abundance of miRNAs.

In the present study, miR-125b, miR-155, miR-200c, miR-205, miR-218, miR-375, and miR-96 were without diagnostic potential. Concordantly, miR-205 had no discriminating capacity between patients with and without positive prostate biopsies in the study by Stephan et al., albeit miR-205 being highly downregulated in PCa tissues [37]. Salido-Guadarrama et al. showed a significant upregulation of miR-200c by 2.6-fold in urinary cells from patients with positive prostate biopsy compared to those with negative biopsy, but they did not further evaluate miR-200c as a diagnostic biomarker [30]. However, it can be assumed that miR-200c might distinguish between patients with and without PCa in their study cohort. Foj et al. and Stuopelytė et al. both showed that miR-375 was significantly upregulated in urinary cells from PCa patients compared to healthy men and BPH patients, respectively [36,41]. Consequently, both studies identified miR-375 as a predictor for PCa (AUC = 0.684–0.797), although they differed distinctly in their selection of study subjects (Table 5). A combination of miR-375 with either miR-21 [36] or miR-148a [41] resulted in a further enhanced diagnostic power (AUC = 0.872 and AUC = 0.785–0.835).

As mentioned before, the selection of patients varied in the discussed studies from including pre-biopsy to pre-RPE men in the PCa group and pre-biopsy to healthy men in the Tf group (Table 5). Furthermore, the number of men in the PCa and Tf group was unequal in some studies [36,40,41,42]. An advantage of our study is the use of post-DRE and pre-biopsy urines, which better reflects the clinical situation of a diagnostic cohort. An ideal biomarker test should be applied before prostate biopsy in order to prevent unnecessary biopsies and thus, limit overdiagnosis and overtreatment. Furthermore, we did not exclude low-risk PCa from the analysis like the study by Salido-Guadarrama et al. did [30]. However, it has to be mentioned that all patients were referred to our department for prostate biopsy due to an elevated serum PSA level ≥4 ng/mL and/or tumor-suspicious DRE. Thus, we are aware of this selection bias. Overall, our approach was most comparable to the studies by Stephan et al. [37], Casanova-Salas et al. [38], and Nayak et al. [39], which used post-DRE and pre-biopsy urines in equally sized PCa and Tf groups. The greatest limitation of our study is the small sample size. Nevertheless, the diagnostic potential of miR-16 and miR-195 remained significant even after adjustment for multiple testing and thus, it would be worthwhile to further investigate both miRNAs as diagnostic biomarkers for PCa. Additionally, only 82% of patients received additional mpMRI before prostate biopsy. Therefore, the pre-biopsy imaging work-up is quite inhomogeneous. Since the main objective of this study was the evaluation of urine-based miRNAs as biomarkers for the detection of PCa within a small study cohort, we decided not to include imaging data in the statistical analyses. However, this should be considered for the validation of this biomarker combination in a larger prospective patient cohort.

Overall, the present findings further highlight the potential of urinary miRNAs, particularly miR-16 and miR-195, as non-invasive biomarkers for PCa diagnostics with high clinical performance. In combination with established PCa markers, such as PSAD, urinary miRNAs could further improve their diagnostic accuracy and thus, help to avoid unnecessary biopsies.

## Figures and Tables

**Figure 1 diagnostics-10-00578-f001:**
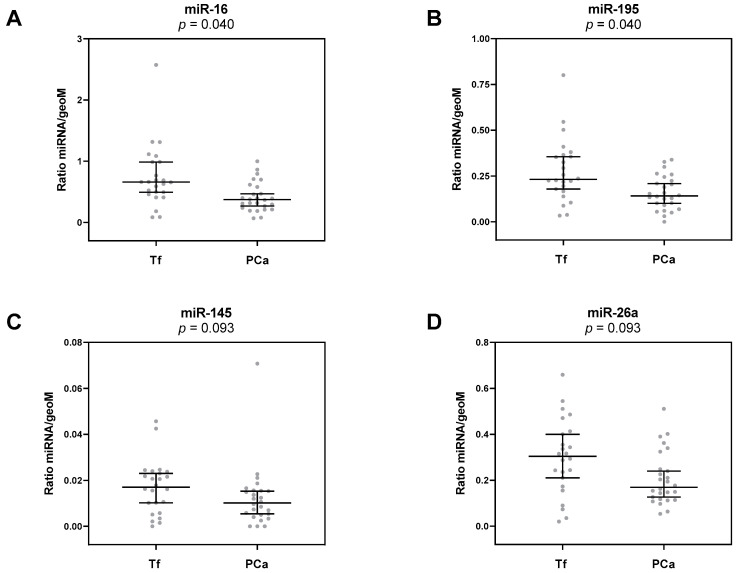
Relative expression levels of (**A**) miR-16, (**B**) miR-195, (**C**) miR-145, and (**D**) miR-26a in urinary sediments from Tf and PCa groups. Depicted are the median relative transcript levels ± 95% CI of the evaluated miRNAs (normalized to the geometric mean of reference RNAs RNU44 and RNU48) in Tf and PCa groups. *p* values were calculated by the Mann–Whitney U test and then corrected for multiple comparisons by the Benjamini–Hochberg method.

**Figure 2 diagnostics-10-00578-f002:**
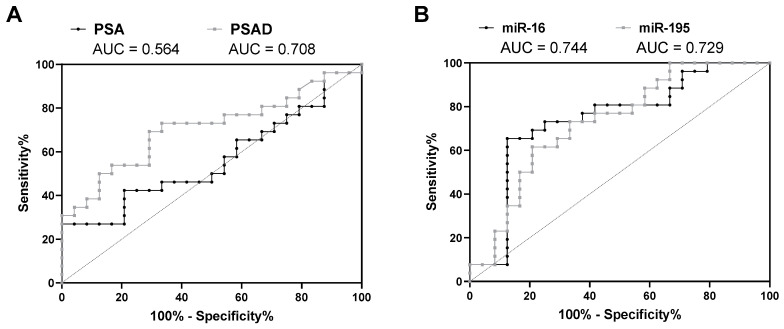
ROC curve analysis of (**A**) serum PSA levels and PSAD as well as of (**B**) miR-16 and miR-195 expression levels in urinary sediments using the whole study cohort (*n* = 50).

**Figure 3 diagnostics-10-00578-f003:**
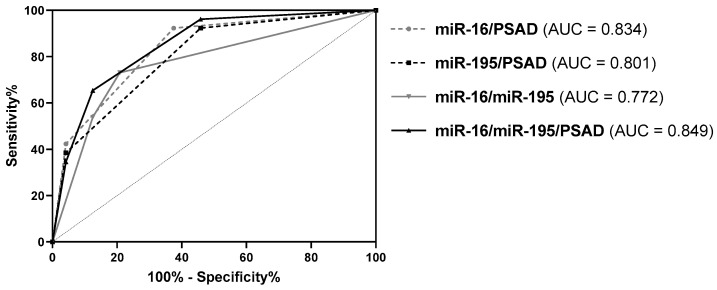
ROC curve analysis of the combinations miR-16/PSAD, miR-195/PSAD, miR-16/miR-195, and miR-16/miR-195/PSAD using the whole study cohort (*n* = 50).

**Figure 4 diagnostics-10-00578-f004:**
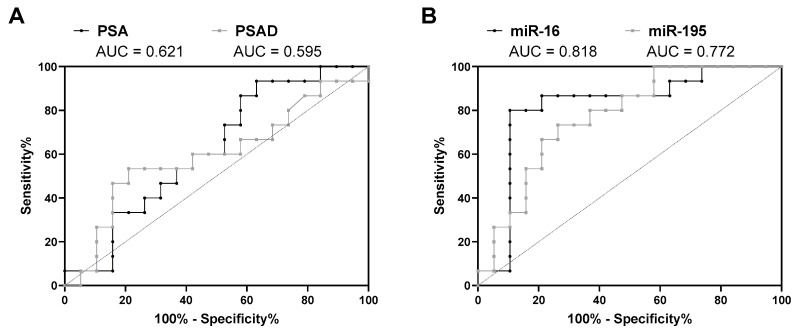
ROC curve analysis of (**A**) serum PSA levels and PSAD as well as of (**B**) miR-16 and miR-195 expression levels in urinary sediments using the sub-cohort of patients with PSA levels ≤ 10 ng/mL (*n* = 34).

**Figure 5 diagnostics-10-00578-f005:**
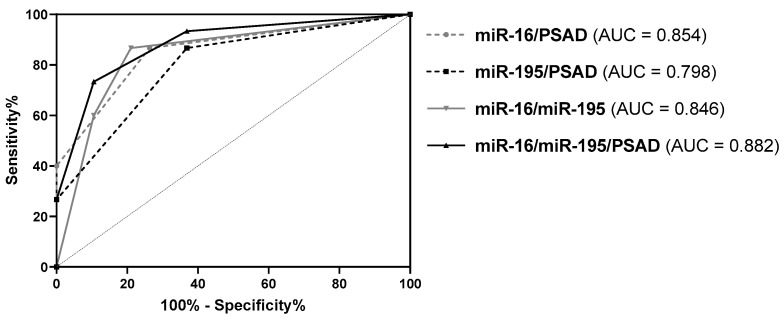
ROC curve analysis of the combinations miR-16/PSAD, miR-195/PSAD, miR-16/miR-195, and miR-16/miR-195/PSAD using the sub-cohort of patients with PSA levels ≤ 10 ng/mL (*n* = 34).

**Table 1 diagnostics-10-00578-t001:** Demographic and clinico-pathological characteristics of patients included in the study.

Parameter	Total Cohort	Tf Group	PCa Group	*p* Value PCa vs. Tf
**Number of Patients**	*n* = 50	*n* = 24	*n* = 26	NA
**Median Age (Years)** (Range)	69.1 (46.8–81.0)	67.4 (46.8–79.4)	71.5 (50.8–81.0)	0.428
**Median PSA Level (ng/mL)** (Range)	7.90 (1.69–50.26)	7.95 (1.81–15.54)	7.86 (1.69–50.26)	0.714
**PSA Level ≤ 10 ng/mL**	*n* = 34	*n* = 19	*n* = 15	NA
**Median PSAD (ng/mL^2^)** (Range)	0.15 (0.03–1.24)	0.12 (0.05–0.34)	0.19 (0.03–1.24)	0.060
**Pre-Biopsy DRE Status** Negative Positive	*n* = 32 (64%) *n* = 18 (36%)	*n* = 19 (79.2%) *n* = 5 (20.8%)	*n* = 13 (50%) *n* = 13 (50%)	0.093
**T Stage** cT1 cT2 cT3 cT4	NA	NA	*n* = 20 (76.9%) *n* = 5 (19.2%) *n* = 0 (0%) *n* = 1 (3.8%)	NA
**N Stage** cN0 cN1 Unknown	NA	NA	*n* = 24 (92.3%) *n* = 1 (3.8%) *n* = 1 (3.8%)	NA
**M Stage** cM0 cM1	NA	NA	*n* = 24 (92.3%) *n* = 2 (7.7%)	NA
**Gleason Grade Group** 1 (GS ≤ 6) 2 (GS = 3 + 4) 3 (GS = 4 + 3) 4 (GS = 8) 5 (GS = 9–10)	NA	NA	*n* = 7 (26.9%) *n* = 9 (34.6%) *n* = 5 (19.2%) *n* = 0 (0%) *n* = 5 (19.2%)	NA

*p* values were calculated by the Mann–Whitney U or chi-squared test and then corrected for multiple comparisons by the Benjamini–Hochberg method. NA: not applicable.

**Table 2 diagnostics-10-00578-t002:** Relative expression levels of miRNAs and respective fold changes in urinary sediments from Tf and PCa groups.

miRNA	Median Relative Expression Level (× 10^−3^)		Fold Change PCa vs. Tf	*p* Value
	in Tf	in PCa		
miR-125b	32.86	30.96	0.94	0.875
miR-145	17.03	10.14	0.60	0.093
miR-155	17.45	21.47	1.23	0.992
miR-16	659.40	373.55	0.57	0.040
miR-195	231.38	141.14	0.61	0.040
miR-200c	689.41	745.59	1.08	0.848
miR-205	36.91	47.88	1.30	0.875
miR-21	579.42	383.47	0.66	0.282
miR-218	2.10	1.86	0.89	0.875
miR-26a	304.50	169.37	0.56	0.093
miR-375	45.93	55.49	1.21	0.944
miR-96	0.28	0.16	0.59	0.365

Depicted are the median relative transcript levels of the evaluated miRNAs (normalized to the geometric mean of reference RNAs RNU44 and RNU48) in Tf and PCa groups as well as the fold change in the PCa samples compared to the Tf group. *p* values were calculated by the Mann–Whitney U test and then corrected for multiple comparisons by the Benjamini–Hochberg method.

**Table 3 diagnostics-10-00578-t003:** Comparison of the diagnostic power of PSAD as well as of miR-16 and miR-195 expression levels in urinary sediments individually and in various combinations using the whole study cohort (*n* = 50).

Parameter	PSAD	miR-16	miR-195	miR-16/PSAD	miR-195/PSAD	miR-16/miR-195	miR-16/miR-195/PSAD
AUC (95% CI)	0.708 (0.562–0.853)	0.744 (0.599–0.888)	0.729 (0.587–0.871)	0.834 (0.721–0.947)	0.801 (0.679–0.924)	0.772 (0.637–0.906)	0.849 (0.741–0.957)
*p* value	0.031	0.012	0.017	< 0.001	0.002	0.005	< 0.001
Cutoff	> 0.15 ^1^	< 0.403	< 0.162	1 of 2	1 of 2	1 of 2	2 of 3
SNS	69.2%	65.4%	61.5%	92.3%	92.3%	73.1%	65.4%
SPC	70.8%	87.5%	79.2%	87.5%	83.3%	79.2%	87.5%
PPV	72.0%	85.0%	76.2%	88.9%	85.7%	79.2%	85.0%
NPV	68.0%	70.0%	65.5%	91.3%	90.9%	73.1%	70.0%
pLR	2.374	5.231	2.954	7.385	5.538	3.508	5.231
nLR	0.434	0.396	0.486	0.088	0.092	0.340	0.396
ACC	70.0%	76.0%	70.0%	90.0%	88.0%	76.0%	76.0%
Ranking	4.	3.	4.	1.	2.	3.	3.

Cutoff values were determined based on the Youden index except for PSAD, where the established cutoff value of 0.15 ng/mL^2^ was used. Ranking of diagnostic performance was achieved by comparing the diagnostic parameters AUC, SNS, SPC, PPV, NPV, pLR, nLR, and ACC. *p* values calculated by the ROC curve analysis were corrected for multiple comparisons by the Benjamini–Hochberg method. ^1^ unit: ng/mL^2.^

**Table 4 diagnostics-10-00578-t004:** Comparison of the diagnostic power of PSAD as well as of miR-16 and miR-195 expression levels in urinary sediments individually and in various combinations using the sub-cohort of patients with PSA levels ≤10 ng/mL (*n* = 34).

Parameter	PSAD	miR-16	miR-195	miR-16/PSAD	miR-195/PSAD	miR-16/miR-195	miR-16/miR-195/PSAD
AUC (95% CI)	0.595 (0.394–0.795)	0.818 (0.659–0.976)	0.772 (0.614–0.930)	0.854 (0.721–0.988)	0.798 (0.646–0.951)	0.846 (0.704–0.987)	0.882 (0.764–1.000)
*p* value	0.524	0.008	0.022	0.004	0.012	0.004	0.004
Cutoff	>0.15 ^1^	<0.403	<0.162	1 of 2	1 of 2	1 of 2	2 of 3
SNS	46.7%	80.0%	66.7%	86.7%	86.7%	86.7%	73.3%
SPC	84.2%	89.5%	78.9%	73.7%	63.2%	78.9%	89.5%
PPV	70.0%	85.7%	71.4%	72.2%	65.0%	76.5%	84.6%
NPV	66.7%	85.0%	75.0%	87.5%	85.7%	88.2%	81.0%
pLR	2.956	7.600	3.167	3.293	2.352	4.117	6.967
nLR	0.633	0.224	0.422	0.181	0.211	0.169	0.298
ACC	67.6%	85.3%	73.5%	79.4%	73.5%	82.4%	82.4%
Ranking	5.	1.	4.	3.	4.	2.	2.

Cutoff values were kept the same as in the ROC curve analysis using the whole study cohort. Ranking of diagnostic performance was achieved by comparing the diagnostic parameters AUC, SNS, SPC, PPV, NPV, pLR, nLR, and ACC. *P* values calculated by the ROC curve analysis were corrected for multiple comparisons by the Benjamini–Hochberg method. ^1^ unit: ng/mL^2^^.^

**Table 5 diagnostics-10-00578-t005:** Overview of selected studies including the present work investigating miRNAs in urinary cells as diagnostic markers for PCa.

Study	Urine Collection	PCa Group	Tf Group	miRNA	Diagnostic Potential ^1^
Present Work	post-DRE pre-biopsy	26 PCa biopsy	24 Tf biopsy	miR-16 miR-195	yes yes
Foj 2017 [36]	post-DRE pre-biopsy (just PCa group)	60 PCa biopsy	10 healthy men	let-7c miR-21 miR-141 miR-214 miR-375 miR-21/375	no yes yes yes yes yes
Salido-Guadarrama 2016 [30]	post-DRE pre-biopsy	73 PCa biopsy (GS ≥ 7)	70 Tf biopsy	miR-100/200b	yes
Stephan 2015 [37]	post-DRE pre-biopsy	38 PCa biopsy	38 Tf biopsy	miR-183 miR-205	no no
Casanova-Salas 2014 [38]	post-DRE pre-biopsy	47 PCa biopsy	45 Tf biopsy	miR-182 miR-187	no yes
Nayak 2020 [39]	post-DRE pre-biopsy	33 PCa biopsy	30 Tf biopsy	miR-182 miR-187	not evaluated
Bryant 2012 [40]	post-DRE post-biopsy	118 PCa biopsy	17 Tf biopsy	miR-107 miR-574-3p	yes yes
Stuopelytė 2016 [42]	post-biopsy & pre-RPE	143 PCa RPE	23 BPH RPE	miR-21 miR-21/19a/19b	yes yes
Stuopelytė 2016 [41]	post-biopsy & pre-RPE (except healthy men)	(a) 143 PCa RPE (b) 72 PCa RPE	(a) 23 BPH RPE (b) 62 healthy men	miR-148a miR-375 miR-148a/375	yes yes yes

^1^ The diagnostic potential was either evaluated by ROC curve analysis or by univariate logistic regression models. RPE: radical prostatectomy.

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
