# Peer review of "Evaluation of MicroRNAs as Non-Invasive Diagnostic Markers in Urinary Cells from Patients with Suspected Prostate Cancer"

_diagnostics, 2020, doi:10.3390/diagnostics10080578_

Round 1

Reviewer 1 Report

In this manuscript, Dr. Angelika Borkowetz and colleagues analyze the expression of 12 miRNAs commonly deregulated in prostate cancer (PCa) in urinary sediments,  obtained post-digital-rectal examination and pre-biopsy in 50 patients with suspected PCa.

The goal is to compare the results with PSA assay, for finding a better method for the diagnosis of Prostate cancer, since PSA generate False results. Results revealed a potential for non-invasive urine-based PCa  detection for miR-16 and miR-195 with greater accuracy than PSA and PSAD.

Unfortunately, the cohort of patients is very small.

50 patients are a good number, but little for more general conclusions.

I think that this manuscript, however, is a good starting point. It is well written, experiments are clear and well performed and statistical analysis is clear.

I have some curiosities:

-have the authors results about Androgen receptor expression (AR) in the prostate cancer samples analyzed?

-have they data about the possible neuroendocrine differentiation of these types of PCa samples? (Recently, it has emerged that NGF signalling is important in PCa and that probably there is a cross-talk between AR and TRK a receptor).

If they have results about these details, they should add them, since they could be helpful in correlating PCa receptors expression to specific diagnostic tests.

Reviewer 2 Report

The authors developed a novel diagnostic method to screen prostate cancer using urine microRNA. Both miR-16 and -195 were inhibited in prostate cancer patients compared to non-cancer patients. With high sensitivity and specificity, combination of PSAD and miR in urine discriminated cancer patients without invasiveness. This study is small sample sized but shows high potential to further develop diagnostic methods in prostate cancer. However, this study lacks information of diagnostic evaluation other than PSA and miRs.

  1. Providing miR in urine can be used as a novel methods for diagnostic tool in prostate cancer, it might be better to remove PSA information and use miR information alone to see whether miR itself can diagnose prostate cancer. That is, subjects in this study have already undergone PSA check and basically PSA>4 were included. This fact is major limitation and if authors cannot assess miR without PSA data they should state and discuss the limitation prudently.
  2. Information of other diagnostic tools, such as trans-rectal US, MRI, PET and other modalities is not described. Readership may hope to know the relationship between multiple use of diagnostic methods with PSA clinically available and miRs.
  3. Non-significant miRs, not -16 and -195, are excessively emphasized in manuscript.
  4. Discussion part is redundant.
